# Vesicant infusates are not associated with ultrasound-guided peripheral intravenous catheter failure: A secondary analysis of existing data

**Amit Bahl**[1]☯*, **Mahmoud Hijazi**[2], **Nai-Wei Chen**[3]☯

1 Department of Emergency Medicine, Beaumont Hospital, Royal Oak, Michigan, United States of America,
2 Oakland University William Beaumont School of Medicine, Rochester, Michigan, United States of America,
3 Department of Biostatistics, Beaumont Hospital, Royal Oak, Michigan, United States of America

☯ These authors contributed equally to this work.
* Amit.bahl@beaumont.edu

## Abstract

**Data Availability Statement:** Due to legal restrictions, data are available from the Beaumont Institutional Data Access/Ethics Committee. Please contact the corresponding author: amit.

### Background

Intravenous vesicants are commonly infused via peripheral intravenous catheters (PIVC) despite guidelines recommending administration via central route. The impact of these medications on PIVC failure is unclear. We aimed to assess dose-related impact of these caustic medications on ultrasound-guided (US) PIVC survivorship.

### Methods

We performed a secondary analysis of a randomized control trial that compared survival of two catheters: a standard long (SL) and an ultra-long (UL) US PIVC. This study involved reviewing and recording all vesicants infusions through the PIVCs. Type and number of vesicants doses were extracted and characterized as one, two or multiple. The most commonly used vesicants were individually categorized for further analysis. The primary outcome was PIVC failure accounting for use and timing of vesicant infusates.

### Results

Between October 2018 and March 2019, 257 subjects were randomized with 131 in the UL group and 126 in the SL group. Vesicants were infused in 96 (37.4%) out of 257 study participants. In multivariable time-dependent extended Cox regression analysis, there was no significant increased risk of failure due to vesicant use [adjusted hazard ratio, aHR 1.71 (95% CI 0.76–1.81) p = 0.477]. The number of vesicant doses was not significantly associated with the increased risk of PIVC failure [(1 vs 0) aHR 1.20 (95% CI 0.71–2.02) p = 0.500], [(2 vs 0) aHR 1.51 (95% CI 0.67–3.43) p = 0.320] and [($\geq$ 3 vs 0) aHR 0.98 (95% CI 0.50–1.92) p = 0.952].

bahl@beaumont.edu or IRB administrative manager: lynne.paul@beaumont.org if you are a researcher that meets criteria for access to confidential data.

**Funding:** The authors received no specific funding for this work.

**Competing interests:** The authors have read the journal's policy and have the following competing interests: AB is a paid consultant for B. Braun. He provides expertise regarding vascular access products and services. All other authors have no relevant disclosures. There are no patents, products in development, or marketed products associated with this research to declare. This does not alter our adherence to PLOS ONE policies on sharing data and materials.

## Conclusion

Vesicant usage did not significantly increase the risk of PIVC failure even when multiple doses were needed in this investigation. Ultrasound-guided PIVCs represent a pragmatic option when vesicant therapy is anticipated. Nevertheless, it is notable that overall PIVC failure rates remain high and other safety events related to vesicant use should be considered when clinicians make vascular access decisions for patients.

## Introduction

With over 300 million used annually in the United States, peripheral intravenous catheters (PIVC) are the most commonly used invasive device in the acute care setting [1,2]. It is estimated that up to 90% of patients require intravenous (IV) access during their hospitalization [3–5]. Unfortunately, PIVCs have high failure rates with up to 63% failing prematurely [3,6]. PIVC failure leads to significant patient harm in the form of repeated insertion attempts, treatment delays, venous depletion, prolongation of hospital stay, psychological and physical stress from needlesticks, and increased rates of nosocomial infections [7–10]. Further, the healthcare costs associated with complications are astounding and the financial burden of premature catheter failure is at a minimum of $1.5 billion nationally [11,12].

Certain intravenous medications categorized as vesicants and irritants are known to cause vein and local tissue injury. The Infusion Nursing Society (INS) created this list based on infusate osmolarity thresholds that increase the likelihood of these complications [13]. These medications can cause local reactions at the level of the vein precipitating vein wall injury, inflammation, thrombus formation, extravasation into the surrounding tissue and potentially lead to tissue necrosis. Premature catheter failure at a minimum is a likely consequence but patients may require additional therapies and even surgical intervention for treatment [14].

It is perceived the impact of these medications on larger proximal veins with higher flow rates and improved hemodilution diminishes the likelihood of these complications. Thus, guidelines support the use of the vesicants and most irritants via central venous lines [15]. Some recent studies have shown that these medications may be safely infused in smaller distal peripheral veins with lower flow rates, particularly many irritants [16–18]. The evidence supporting the delivery of vesicant infusates via PIVCs is limited and has mixed results regarding patient safety [19]. As most studies have focused on complications, specifically extravasation, there is a paucity of evidence related to these caustic infusions and the impact on catheter survivorship. Further, the cumulative impact of multiple doses of vesicants on catheter survival and complications has not been evaluated.

We aimed to explore the impact of vesicant administration on IV catheter survival and complications for PIVCs as well as assess for differences in survival when catheters were dichotomized by length.

## Methods

### Study sample

The study was conducted at a large 1100 bed tertiary care center with an annual emergency department census of approximately 130,000 visits. We performed a secondary analysis of an existing single site, randomized control trial that directly compared two catheters: (1) a standard long (SL) 20-gauge 4.78 cm Becton Dickinson (BD) Insyte™ Autoguard™ IV catheter and

(2) an ultra-long (UL) 20-gauge 6.35 cm B. Braun Introcan Safety® IV catheter (Clinical-Trials.gov: NCT03655106). The primary study was approved for adult patients at least 18 years of age with self-reported difficult vascular access (DVA) and at least one of the following: history of requiring 2 or more intravenous attempts on a previous visit, previous requirement for a rescue catheter (ultrasonographically guided intravenous catheter, midline catheter, or central venous access), end-stage renal disease and receiving dialysis, injection drug use, or sickle cell disease. Patients were excluded if they were previously enrolled, withdrew from the study, or presented when trained intravenous line inserters were unavailable. Finally, 257 participants were included for analysis. This secondary analysis from a primary study was approved by the Institutional Review Board (IRB) of Beaumont Health.

The research team assessed catheters for functionality at the time of insertion and daily for the life of the catheter. A functional catheter could be flushed with 5 mL of normal saline without resistance and without complication. Research staff noted whether the study catheter survived to completion of therapy or failed prematurely. If the catheter failed before the follow-up, the date, time, and reason for failure were collected from the patient's medical record. If a patient was discharged before their follow-up, the line was presumed functional unless stated otherwise and the line removal date and time were collected from the patient's chart. Follow-up visits included reviewing and recording all medications administered via the intravenous lines. Complications were identified during daily site assessments and chart reviews and included: phlebitis, infiltration, dislodgment, occlusion, and leaking.

### Independent variables

Three scenarios related to vesicant infusates were considered on analysis. (1) Specifically, vesicants and irritant infusates as recognized by the INS were noted (S1 Appendix with list of all infusates used in study trial) and use of vesicants was characterized as yes/no. (2) Number of vesicants/irritants doses were extracted and characterized as one, two, or multiple. (3) The most commonly used vesicants were further individually categorized for further analysis and included: vancomycin, dextrose 50% solutions, and intravenous contrast for computed tomography.

### Outcome variable

The primary outcome was the premature catheter failure of US PIVC.

### Primary data analysis

Continuous data were shown as means (standard deviations; SD) and categorical data as frequencies (percentages). Clinical characteristics were summarized by use of vesicants and irritant infusates and compared by using t-test and $\chi^2$ test for continuous and categorical variables, respectively.

To explore the association between vesicant infusates and PIVC failure, the effect of vesicants on catheter failure was assessed with time-dependent extended survival analysis, which accounted for timing of vesicant administration. The median survival time of catheters by vesicant infusates was estimated in the modified Kaplan-Meier (K-M) survival curves (e.g., stsplit in statistical software, Stata). To examine if the effect of vesicants on catheter failure was dependent on the length of catheters (UL and SL), we initially tested the interaction term of vesicants infusates and length of catheters in Cox regression model and results indicated that the interaction was not statistically significant (ie, the effect of vesicants on catheter failure did not statistically significantly depend on length of catheters.) (S2 Appendix). Hence, in multivariable Cox regression, we refitted models without the interaction term of vesicant infusates

and length of catheters to assess the effect of vesicants infusates adjusted for length of catheters only and other clinical characteristics of patients pre-specified in the original trial study [20], including age, sex, history of end-stage renal disease, body mass index, systolic blood pressure and pulse rates. We report the hazard ratio (HR) with 95% confidence interval (CI). All tests with $p < 0.05$ were considered to indicate statistical significance. All statistical analyses were performed with Stata 15.1 (StataCorp) and SAS v9.4 (SAS Institute, Inc., Cary, NC).

## Results

Between October 2018 and March 2019, 270 patients were randomized to ultra-long (135) and standard long (135) groups. After some exclusions, the final data set included 257 subjects with 131 in the UL group and 126 in the SL group. The average age was 59.2 (SD 18.4) with 71.6% female gender. Average BMI was 31.7 (SD 10.0) and 16.7% had end-stage renal disease. The average vein diameter was 0.3 cm (SD 0.1) and vein depth was 1.0 cm (SD 0.3). Vesicants were infused in 96 (37.4%) out of 257 study participants. (Table 1) Forty-four (45.8%) of catheters with vesicants failed while 50 (31.1%) of catheters without vesicants failed (p = 0.017). In the UL group, 18/41 (43.9%) failed with vesicant/irritant used compared to 23/90 (25.6%) without vesicant/irritant use. In the SL group, 26/55 (47.3%) failed with vesicant/irritant use compared to 27/71 (38.0%) without vesicant/irritant use. There was no difference in the occurrence of PIVC failure by use of vesicant/irritant across US PIVC groups (Breslow-Day-Tarone test, p = 0.409) (Table 2).

In the modified K-M curves, results indicated that there was no significant difference in median survival duration in PIVC by use of vesicant/irritant [(vesicant/irritant use vs no vesicant/irritant use), 6 vs 7 days p = 0.18]. (Fig 1) In univariable time-dependent Cox regression analysis, there was no significant increased risk of failure due to vesicant use [HR 1.31 (95% CI 0.87–1.98) p = 0.195]. Similarly, in multivariable Cox model, there was no significant impact of caustic infusates on PIVC failure [adjusted hazard ratio, aHR 1.17 (95% CI 0.76–1.81) p = 0.477] (Table 3).

When assessing the impact of number of doses on failure, there was no statistically significant association between catheter failure and vesicant infusate with one dose [(1 vs 0) aHR 1.20 (95% CI 0.71–2.02) p = 0.500]. Similarly, when 2 doses and $\geq$ 3 doses were infused, there were no statistically significant effect on the risk of IV catheter failure, respectively, [(2 vs 0) aHR 1.51 (95% CI 0.67–3.43) p = 0.320; ($\geq$ 3 vs 0) aHR 0.98 (95% CI 0.50–1.92) p = 0.952] (Table 4).

The three most commonly used caustic infusates were IV contrast (51), vancomycin (28), and dextrose containing solutions (14). When considering these medication types, each of the vesicant infusates had no significant effect on the risk of PIVC failure, respectively (Table 5).

Phlebitis was noted on clinical assessment in 14 cases with 3 cases in the UL group and 11 cases in the SL group. No cases of phlebitis in the UL group involved vesicant/irritant use. 7 cases in the SL group were associated with vesicant use and 4 cases were not associated with vesicant use (S3 Appendix).

## Limitations

Our study had several limitations. First, this investigation was conducted at a single site in a large academic suburban tertiary care center and results may not be generalizable to other settings and populations. Second, in some scenarios patients had multiple intravenous lines and it was not always clear what access point was used for the caustic infusion. Third, a number or medications were grouped together for the majority of the analysis and the impact on catheter survival and complications may be different for individual therapies. Specifically, a small

**Table 1. Patient and intravenous-related characteristics stratified by use of vesicant and irritant infusates.**

| Variables | All | | | Vesicant and Irritant Infusates | | | | | p value |
|---|---|---|---|---|---|---|---|---|---|
| | | | | Yes | | No | | | |
| N | 257 | | | 96 | | 161 | | | |
| Total dose | | | | | | | | | – |
| 0 | 161 | (62.7) | | – | | 161 | | (100) | |
| 1 | 55 | (21.4) | | 55 | (57.3) | – | | | |
| 2 | 14 | (5.4) | | 14 | (14.6) | – | | | |
| ≥ 3 | 27 | (10.5) | | 27 | (28.1) | – | | | |
| **Patient characteristics** | | | | | | | | | |
| Age, years, mean (SD) | 59.2 | (18.4) | | 59.1 | (17.6) | 59.3 | (18.8) | | 0.944 |
| Sex, No. (%) | | | | | | | | | |
| Male | 73 | (28.4) | | 26 | (27.1) | 47 | (29.2) | | 0.717 |
| Female | 184 | (71.6) | | 70 | (72.9) | 114 | (70.8) | | |
| ESRD, No. (%) | | | | | | | | | |
| No | 214 | (83.3) | | 79 | (82.3) | 135 | (83.9) | | 0.746 |
| Yes | 43 | (16.7) | | 17 | (17.7) | 26 | (16.1) | | |
| BMI, kg/m$^2$, mean (SD) | 31.7 | (10.0) | | 30.8 | (9.4) | 32.3 | (10.3) | | 0.264 |
| Systolic blood pressure, mmHg, mean (SD) | 141.1 | (27.9) | | 139.8 | (27.2) | 141.8 | (28.4) | | 0.585 |
| Diastolic blood pressure, mmHg, mean (SD) | 75.2 | (14.9) | | 75.4 | (15.0) | 75.1 | (14.9) | | 0.853 |
| Pulse rate, bpm, mean (SD) | 88.9 | (19.3) | | 90.8 | (18.2) | 87.9 | (19.9) | | 0.246 |
| **Intravenous characteristics** | | | | | | | | | |
| Ultrasound-guided PIVC, No. (%) | | | | | | | | | |
| Standard Long, SL | 126 | (49.0) | | 55 | (57.3) | 71 | (44.1) | | 0.041 |
| Ultra Long, UL | 131 | (51.0) | | 41 | (42.7) | 90 | (55.9) | | |
| Depth of vein, cm, mean (SD) | 1.0 | (0.3) | | 1.0 | (0.3) | 1.0 | (0.3) | | 0.989 |
| Diameter of vein, cm, mean (SD) | 0.3 | (0.1) | | 0.3 | (0.1) | 0.4 | (0.1) | | 0.119 |
| Duration of line function, days, mean (SD) | 2.6 | (3.5) | | 3.5 | (4.6) | 2.1 | (2.4) | | 0.002 |

SD, standard deviation; ESRD, end-stage renal disease; BMI, body mass index; PIVC, peripheral intravenous catheter.

number of patients received vasopressors via peripheral catheter, and the results may be limited for this class of therapies. Fourth, 96% of all catheters in this study were placed using ultrasound-guided technique in the upper arm and the risk of failure and complications due to vesicant infusion may not be the same as traditionally placed catheters or with ultrasound-guidance for alternate locations. As there is limited existing literature for all peripheral catheters on this topic, we could not exclusively focus on ultrasound-guided insertions in the discussion section. Finally, as many catheters failed outside of the daily assessment time periods,

**Table 2. Summary on intravenous line failure.**

| | Ultra Long PIVC (UL) | | Standard Long PIVC (SL) | |
|---|---|---|---|---|
| | Vesicant and Irritant Infusates | | Vesicant and Irritant Infusates | |
| Line function | Yes | No | Yes | No |
| Failure | 18 | 23 | 26 | 27 |
| Not failure | 23 | 67 | 29 | 44 |

PIVC, peripheral intravenous catheter.

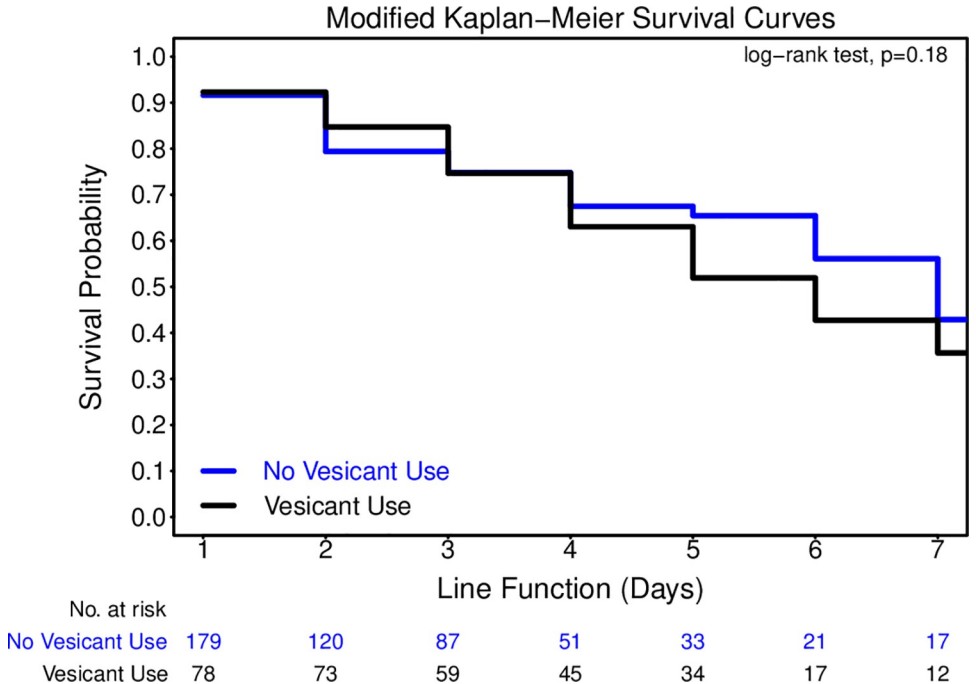

**Fig 1. Modified Kaplan-Meier survival curve estimates by status of vesicant use for PIVC survival.**

there was heavy reliance and potential inaccuracies on nursing documentation regarding line complications and etiology of failure.

## Discussion

While caustic infusates are commonly delivered via peripheral catheters, the impact of this practice on catheter failure is largely unknown. In fact, as far as we are aware, this is the first investigation that has evaluated vesicant use in peripheral catheters with the primary focus on catheter failure. While on the surface results indicate that any use of caustic infusates may increase the likelihood of PIVC failure, when time of administration was considered in the survival analysis, we determined that caustic infusates did not increase the risk of PIVC failure. When assessing catheter failure by catheter length, the longer 2.5 inch catheter was less likely to experience failure than the shorter 1.88 inch catheter. While the impact of IV length on

**Table 3. Effects of any use of vesicant/irritant infusates on intravenous line failure.**

| Effects[¶] | Model 1[§] | | Model 2[‡] | |
|---|---|---|---|---|
| | Unadjusted HR (95% CI) | *p* value | Adjusted HR (95% CI) | *p* value |
| Vesicant/Irritant Infusates (yes *vs* no) | 1.31 (0.87–1.98) | 0.195 | 1.17 (0.76–1.81) | 0.477 |
| Ultrasound-guided PIVC (UL *vs* SL) | | | 0.45 (0.29–0.72) | 0.001 |

*PIVC*, peripheral intravenous catheter; *UL*, ultra long; *SL*, standard long; *BMI*, body mass index; *ESRD*, end-stage renal disease; *SBP*, systolic blood pressure; *HR*, hazard ratio; *CI*, confidence interval.

[¶] Use of vesicant/irritant infusates was time-dependent variable.

[§] Model 1 only included use of vesicant/irritant infusates.

[‡] Model 2 included use of vesicant/irritant infusates, adjusted for PIVC type and other clinical characteristics including age, sex, BMI, ESRD, SBP, pulse rate, and depth of vein (S2 Appendix). Only the adjusted effects of use of vesicant/irritant infusates and PIVC type were shown.

**Table 4. Effects of total dose of vesicant/irritant infusates on intravenous line failure.**

| Effects[¶] | Model 1[§] | | Model 2[‡] | |
|---|---|---|---|---|
| | Unadjusted HR (95% CI) | p value | Adjusted HR (95% CI) | p value |
| Vesicant/Irritant Infusates | | | | |
| Total dose (1 vs 0) | 1.31 (0.80–2.15) | 0.284 | 1.20 (0.71–2.02) | 0.500 |
| Total dose (2 vs 0) | 1.69 (0.76–3.75) | 0.195 | 1.51 (0.67–3.43) | 0.320 |
| Total dose (≥3 vs 0) | 1.16 (0.61–2.20) | 0.652 | 0.98 (0.50–1.92) | 0.952 |
| Ultrasound-guided PIVC (UL vs SL) | | | 0.45 (0.29–0.72) | 0.001 |

PIVC, peripheral intravenous catheter; UL, ultra long; SL, standard long; BMI, body mass index; ESRD, end-stage renal disease; SBP, systolic blood pressure; HR, hazard ratio; CI, confidence interval.

[¶] Total dose of vesicant/irritant infusates was time-dependent variable.

[§] Model 1 only included total dose of vesicant/irritant infusates.

[‡] Model 2 included total dose of vesicant/irritant infusates, adjusted for PIVC type and other clinical characteristics including age, sex, BMI, ESRD, SBP, pulse rate, and depth of vein. Only the adjusted effects of total dose of vesicant/irritant infusates and PIVC type were shown.

catheter failure has not previously been specifically evaluated in the setting of vesicant usage, most studies only describe short-term (less than 24 hours) infusions of vesicants with standard length PIVCs compared to longer infusion durations with longer peripheral midline catheters suggesting a benefit of longer catheters [21,22].

Interestingly, the risk of catheter failure was also not significantly impacted by the number of caustic infusate doses administered. This dose-dependent relationship of caustic infusates and complications/failure hasn't been previously described in the current literature. In regards to specific infusates, vancomycin, non-ionic Iodine-based contrast, and dextrose 50% were the most commonly used caustic infusates in our cohort and did not significantly enhance the risk of early line failure in both catheter types. The use of vancomycin via peripheral IV route is a heavily debated topic with no clear recommendation. Mowry described increased rates of phlebitis, thrombosis, and extravasation when vancomycin is administered peripherally [15]. On the other hand, Caparas et al. found that midlines had a slightly higher rate of complications than PICC lines for short-term vancomycin treatment (less than 6 days), but found no difference in the rate of phlebitis or thrombosis [23]. Based on our results, the risk of premature PIVC failure is insignificant with these vesicants and US-guided PIVCs may be considered even when multiple doses are required.

In addition to poor catheter survival outcomes, the safety of delivering caustic infusions via PIVCs also needs consideration. Current safety guidelines on use of peripheral catheters for vesicants and irritants are vague and without clear direction. The Infusion Nursing Society

**Table 5. Effects of each specific vesicant/irritant infusates on intravenous line failure in separate analyses.**

| Effects[¶] | Specific Type of Medication[§] | | | | | |
|---|---|---|---|---|---|---|
| | IV Contrast | | Vancomycin | | Dextrose 50% | |
| | Adjusted HR (95% CI) | p value | Adjusted HR (95% CI) | p value | Adjusted HR (95% CI) | p value |
| Vesicant/Irritant Infusates (yes vs no) | 1.32 (0.79–2.21) | 0.295 | 1.07 (0.60–1.90) | 0.824 | 0.78 (0.33–1.80) | 0.555 |
| Ultrasound-guided PIVC (UL vs SL) | 0.45 (0.29–0.72) | 0.001 | 0.44 (0.28–0.70) | 0.001 | 0.44 (0.28–0.69) | < 0.001 |

PIVC, peripheral intravenous catheter; UL, ultra long; SL, standard long; HR, hazard ratio; CI, confidence interval.

[¶] Use of specific vesicant/irritant infusates was time-dependent variable in each separate analysis.

[§] Each regression analysis included a specific vesicant/irritant infusates, PIVC type, and other clinical characteristics including age, sex, BMI, ESRD, SBP, pulse rate, and depth of vein. Only the adjusted effects of vesicant/irritant infusates and PIVC type were shown.

standards of practice recommends cautious infusion of vesicants and irritants via peripheral route with a hard-stop only for continuous vesicant therapy [13]. The MAGIC guidelines on the other hand do not recommend the peripheral route for non-peripherally compatible infusates [24]. The limited existing literature also provides little concrete evidence regarding patient safety with this practice and is uniquely focused on the complications of extravasation and local tissue injury. In our study sample, the risk of extravasation was not insignificant. While there were 10 extravasation events, fortunately no patients required invasive therapies for treatment. Similarly, the evidence suggests extravasation events in PIVCs are not an insignificant occurrence. A systematic review assessing the safety of PIVCs for vasopressors included 1382 patients found the incidence of extravasation was 3.4% (ranging from 2.3% to 5.5%) with 5.5% in the only prospective study included in the analysis [21,25,26]. While these rates may seem low, these adverse events are still substantially more common in PIVCs compared to central venous catheters CVCs. In one large systematic review, authors found 325 distinct events of local tissue injury and/or extravasation with 318 (97.8%) events occurring in the PIVC group compare to just 7 (2.2%) events in the CVC group [1]. Notably, data exploring other complications such as phlebitis, thrombosis, and infection when vesicants are used via peripherals is sparse. To gain a more complete perspective into the safety profile of vesicant use via US-guided PIVCs, further research is required.

Vesicant usage does not significantly increase the risk of PIVC failure even when multiple doses are needed. Ultrasound-guided PIVCs represent a pragmatic option when vesicant therapy is anticipated. Nevertheless, it is notable that overall PIVC failure rates remain high and other safety events related to vesicant use should be considered when clinicians make vascular access decisions for patients.

## Supporting information

**S1 Appendix. List of medication use in study.**
(DOCX)

**S2 Appendix. Effects of variables on multivariable Cox regression models.**
(DOCX)

**S3 Appendix. Causes of intravenous line removal by vesicants and irritant infusates and PIVC type.**
(DOCX)

## Author Contributions

**Conceptualization:** Amit Bahl, Nai-Wei Chen.

**Data curation:** Amit Bahl, Mahmoud Hijazi, Nai-Wei Chen.

**Formal analysis:** Nai-Wei Chen.

**Investigation:** Amit Bahl.

**Methodology:** Amit Bahl, Nai-Wei Chen.

**Project administration:** Amit Bahl.

**Software:** Nai-Wei Chen.

**Supervision:** Amit Bahl.

**Validation:** Amit Bahl, Nai-Wei Chen.

**Writing – original draft:** Amit Bahl, Mahmoud Hijazi, Nai-Wei Chen.

**Writing – review & editing:** Amit Bahl, Mahmoud Hijazi, Nai-Wei Chen.

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
