## [Decision Letter · Decision Letter 0]

19 Oct 2021

PONE-D-21-15885Vesicant infusates are not associated with ultrasound-guided peripheral intravenous catheter failure: A time-dependent extended analysis of a randomized controlled trialPLOS ONE

Dear Dr. Bahl,

Thank you for submitting your manuscript to PLOS ONE. After careful consideration, we feel that it has merit but does not fully meet PLOS ONE’s publication criteria as it currently stands. Therefore, we invite you to submit a revised version of the manuscript that addresses the points raised during the review process.

The manuscript has been evaluated by three reviewers, and their comments are available below.

The reviewers have raised a number of concerns that need attention. They raise questions over the presentation of statistics and statistical analyses, as well as some other minor comments.

Could you please revise the manuscript to carefully address the concerns raised?

We look forward to receiving your revised manuscript.

Kind regards,

Sebastian Shepherd

Associate Editor

PLOS ONE

Journal Requirements:

2. Thank you for stating the following in the Competing Interests/Financial Disclosure*  section:

“I have read the journal's policy and the authors of this manuscript have the following competing interests: AB is a paid consultant for B. Braun. He provides expertise regarding vascular access products and services. All other authors have no relevant disclosures.”

We note that one or more of the authors are employed by a commercial company: B. Braun

3. Please include a caption for figure 1.

4. Please include your tables as part of your main manuscript and remove the individual files. Please note that supplementary tables (should remain/ be uploaded) as separate "supporting information" files.

Reviewers' comments:

Reviewer's Responses to Questions

**Comments to the Author**

1. Is the manuscript technically sound, and do the data support the conclusions?

Reviewer #1: Partly

Reviewer #2: Yes

Reviewer #3: Yes

2. Has the statistical analysis been performed appropriately and rigorously? 

Reviewer #1: No

Reviewer #2: Yes

Reviewer #3: Yes

3. Have the authors made all data underlying the findings in their manuscript fully available?

Reviewer #1: No

Reviewer #2: No

Reviewer #3: No

4. Is the manuscript presented in an intelligible fashion and written in standard English?

Reviewer #1: Yes

Reviewer #2: Yes

Reviewer #3: Yes

5. Review Comments to the Author

Reviewer #1: PONE-D-21-15885: statistical review

SUMMARY. This is a secondary analysis of an existing randomized control trial. It estimates the influence of vesicant infusates on peripheral intravenous catheters (PIVC) failure. The core statistical analysis relies on a Cox regression model where time up to PIVC failure is the dependent variable and use of vesicant infusates is the principal time-varying covariate. Although the results are sound and the statistical methods seem generally correct, I list below some points that need clarification.

SPECIFIC POINTS

1. I didn’t find any information about the censored cases. How many censored cases do we have for each catheter type? Can we assume that censoring is not informative?

2. Table 1 displays a significant interaction between catheter type and use of infusates (p=0.041) that has not been discussed. How do the authors interpret this interaction?

3. The material relating to the Cox regression results is not fully clear. I’d welcome two tables of results, which respectively display both the Cox regression with all covariates plus the interaction term of vesicant infusates and catheter type, and the Cox regression with all the covariates but without the interaction term. It is important to see whether the covariate effects change significantly when the interaction effect is removed.

4. Cox regression models are capable of handling time-varying covariates by organizing the data as counting processes (i.e. multiple records for each subject that reflect the change of status). I don’t fully understand why the preliminary analysis that involves the Simon-Makuch method and the Mantel-Byar test is really needed here. Please clarify. In particular, I suspect that Mantel-Byar test is equivalent to the ordinary log-rank test when the data are arranged as counting processes…

Reviewer #2: Congratulations on your excellent study and clearly written manuscript. This subject area is one of great interest. While I appreciated the results reflecting extravasation incidence and non-serious outcomes, I would have liked to see a bit more data and explanation on the reasons for PIVC failure within the groups.

Reviewer #3: The authors should be congratulated, it is a well written, prepared, and presented paper, however I do have some comments.

Title: "Vesicant infusates are not associated with ultrasound-guided peripheral intravenous catheter failure: A time-dependent extended analysis of a randomized controlled trial" not sure what time-dependent extended analysis means, perhaps say 'secondary analysis'. Using failure as outcome should imply a time-to-event analysis. It may not be appropriate to mention RCT in the title, since this study was not randomised on vesicant/non-vesicant drugs. It's a secondary analysis of a dataset that happens to come from an RCT.

"Further, the healthcare costs associated with complications are astounding and the financial burden of premature catheter failure is at a minimum of $1.5 billion nationally.7,8" Perhaps also mention the impact on the patient.

Methods:

Need to mention the setting, e.g. name of hospital.

Need to specify the definition of failure, 5mL flushing was mentioned, but the determination of failure based on other signs/symptoms (e.g. pain) should be detailed. These are listed in the supplement but should be in the main text.

Over-fitting of the Cox model could be a concern at this sample size with this many co-variables, especially if the survival analysis is split. May not need to adjust for all these co-variables. Variable selection rationale should be detailed.

Results:

KM plot: should be truncated at around 7 days (remaining data are practically meaningless)

Table 1. What is 'line failure'? Is this the primary outcome? Should be moved to Table 3. Table 2 and 3 should be swapped?

Table 2. What would be the result if the vesicant variable is treated as dichotomous (not time-dependent)? A simple Cox regression?

Cannot see the descriptive statistics of the number of vesicant doses (0/1/2/3).

Discussion:

How do you explain that the effect of vesicant in Table 1 was significant but not significant in the advanced analyses? Should be in Discussion.

Please check this research poster as it may be relevant: https://metronorth.health.qld.gov.au/hhps-2020/CLIN-0111.pdf

6. PLOS authors have the option to publish the peer review history of their article (what does this mean?). If published, this will include your full peer review and any attached files.

Reviewer #1: No

Reviewer #2: **Yes: **Nancy L Moureau

Reviewer #3: No

---

## [Author Response · Author response to Decision Letter 0]

11 Nov 2021

All responses to editor/reviewer comments are detailed in the author response letter. Thank you for the continued consideration.

---

## [Decision Letter · Decision Letter 1]

20 Dec 2021

PONE-D-21-15885R1Vesicant infusates are not associated with ultrasound-guided peripheral intravenous catheter failure: A secondary analysis of existing dataPLOS ONE

Dear Dr. Bahl,

Thank you for submitting your manuscript to PLOS ONE. The reviewers are happy with the revised version of the manuscript but have pointed our some minor concerns that have to be addressed.

We look forward to receiving your revised manuscript.

Kind regards,

Miquel Vall-llosera Camps

Senior Editor

PLOS ONE

Journal Requirements:

Reviewers' comments:

Reviewer's Responses to Questions

**Comments to the Author**

1. If the authors have adequately addressed your comments raised in a previous round of review and you feel that this manuscript is now acceptable for publication, you may indicate that here to bypass the “Comments to the Author” section, enter your conflict of interest statement in the “Confidential to Editor” section, and submit your "Accept" recommendation.

Reviewer #1: All comments have been addressed

Reviewer #2: All comments have been addressed

Reviewer #3: All comments have been addressed

2. Is the manuscript technically sound, and do the data support the conclusions?

Reviewer #1: (No Response)

Reviewer #2: Yes

Reviewer #3: Yes

3. Has the statistical analysis been performed appropriately and rigorously? 

Reviewer #1: (No Response)

Reviewer #2: Yes

Reviewer #3: Yes

4. Have the authors made all data underlying the findings in their manuscript fully available?

Reviewer #1: (No Response)

Reviewer #2: Yes

Reviewer #3: No

5. Is the manuscript presented in an intelligible fashion and written in standard English?

Reviewer #1: (No Response)

Reviewer #2: Yes

Reviewer #3: Yes

6. Review Comments to the Author

Reviewer #1: (No Response)

Reviewer #2: One correction: the reference to INS Standards is not correct

13 Gorski LA. The 2016 infusion therapy standards of practice. Home Healthcare Now. 2017;35(1):10–8.

The correct reference:

Gorski L, Hadaway L, Hagle M, Broadhurst D, Clare S, Kleidon T, Meyer B, Nickel B, Rowley S, Sharpe E, Alexander M. (2021) Infusion Therapy Standards of Practice, 8th Edition. Journal of Infusion Nursing. 2021;44(1S):S1-S224. doi:10.1097/NAN.0000000000000396

Reviewer #3: Thank you for addressing the points raised by the reviewers.

In the Abstract/Conclusions, the first sentence should be in past tense and words "in this study" or similar should be added.

7. PLOS authors have the option to publish the peer review history of their article (what does this mean?). If published, this will include your full peer review and any attached files.

Reviewer #1: No

Reviewer #2: No

Reviewer #3: No

---

## [Author Response · Author response to Decision Letter 1]

20 Dec 2021

please see author response letter

---

## [Editor Report · Decision Letter 2]

6 Jan 2022

Vesicant infusates are not associated with ultrasound-guided peripheral intravenous catheter failure: A secondary analysis of existing data

PONE-D-21-15885R2

Dear Dr. Bahl,

We’re pleased to inform you that your manuscript has been judged scientifically suitable for publication and will be formally accepted for publication once it meets all outstanding technical requirements.

Kind regards,

Miquel Vall-llosera Camps

Senior Editor

PLOS ONE

---

## [Editor Report · Acceptance letter]

17 Jan 2022

PONE-D-21-15885R2 

Vesicant infusates are not associated with ultrasound-guided peripheral intravenous catheter failure: A secondary analysis of existing data 

Dear Dr. Bahl:

I'm pleased to inform you that your manuscript has been deemed suitable for publication in PLOS ONE. Congratulations! Your manuscript is now with our production department. 

Kind regards, 

on behalf of

Dr. Miquel Vall-llosera Camps 

Staff Editor

PLOS ONE